# Dietary Fats and Cardio-Metabolic Outcomes in a Cohort of Italian Adults

**DOI:** 10.3390/nu14204294

**Published:** 2022-10-14

**Authors:** Walter Currenti, Justyna Godos, Amer M. Alanazi, Giuseppe Grosso, Raffaele Ivan Cincione, Sandro La Vignera, Silvio Buscemi, Fabio Galvano

**Affiliations:** 1Department of Biomedical and Biotechnological Sciences, University of Catania, 95123 Catania, Italy; 2Department of Pharmaceutical Chemistry, College of Pharmacy, King Saud University, P.O. Box 2457, Riyadh 11451, Saudi Arabia; 3Department of Clinical and Experimental Medicine, University of Foggia, 71122 Foggia, Italy; 4Department of Clinical and Experimental Medicine, University of Catania, 95123 Catania, Italy; 5Department of Health Promotion, Maternal and Child Health, Internal and Specialty Medicine of Excellence (PROMISE), University of Palermo, 90127 Palermo, Italy

**Keywords:** fat, dietary fats, saturated fats, polyunsaturated fats, monounsaturated fats, short-chain fatty acids

## Abstract

Background: Dietary fats, and especially saturated fatty acid (SFA), have been blamed for being the culprit in the dramatic increase in obesity and its associated diseases. However multiple systematic reviews and recent meta-analyses do not support the association between SFA and cardiovascular diseases. Thus, the objective of this study was to test whether specific types and subtypes of dietary fats are associated with metabolic outcomes in a cohort of Italian adults. Methods: Nutritional and demographic data of 1936 adults living in the south of Italy were examined. Food frequency questionnaires (FFQs) were administered to assess the intake of total dietary fat and each specific class of dietary fat, such as SFA, monounsaturated fatty acid (MUFA) and polyunsaturated fatty acid (PUFA). The intake of fatty acids was also examined according to the carbon-chain length of each individual class. Cases of hypertension, type-2 diabetes and dyslipidemias were collected from previous doctor-confirmed diagnosis records (or direct measurement of blood pressure). Results: After adjustment for potential confounding factors, individuals reporting higher intakes of total and saturated fats were associated with lower likelihood of having hypertension (odds ratio (OR) = 0.57, 95% CI: 0.35, 0.91 and OR = 0.55, 95% CI: 0.34, 0.89, respectively). Moreover, higher intake of short-chain saturated fatty acids (SCSFAs) and medium-chain saturated fatty acids (MCSFAs) was inversely associated with dyslipidemia and diabetes (OR = 0.43, 95% CI: 0.23, 0.82 and OR = 0.25, 95% CI: 0.09, 0.72, respectively). Among MUFAs, C18:1 was inversely associated with hypertension and diabetes (OR = 0.52, 95% CI: 0.30, 0.92 and OR = 0.21, 95% CI: 0.07, 0.67, respectively), while C14:1 intake was inversely associated only with hypertension (OR = 0.57, 95% CI: 0.37, 0.88). In contrast, C20:1 intake was associated with dyslipidemia (OR = 3.35, 95% CI: 1.33, 8.42). Regarding PUFA, C18:2 and 20:5 were inversely associated with hypertension (OR = 0.33, 95% CI: 0.18, 0.60 and OR = 0.30, 95% CI: 0.10, 0.89, respectively). Conclusions: The consumption of SFA does not seem to be harmful to cardio-metabolic health and, on the contrary, SCSFA may exert beneficial effects. Further studies are needed to clearly validate the results of the present study.

## 1. Introduction

Noncommunicable diseases (NCDs), such as cardio-metabolic disorders and cancer, are currently the main causes of global mortality, representing 71% of all deaths in the world [1]. Recent evidence shows that the major risk factor for these conditions is chronic subclinical low-grade inflammation [2], which is usually determined by physical inactivity, tobacco, pollution, sleep alterations, dysbiosis, and infections [3]. Among the triggers for inflammation, poor diet and excessive weight also play important roles [4]. The growth in the volume and number of adipocytes due to an excessive caloric intake leads to an increase in monocyte adhesion and recruitment to adipose tissue. Macrophages in adipocytes show many pro-inflammatory receptors, such as tumor necrosis factor receptors (TNFRs), Toll-like receptors (TLRs) and interleukin-1 receptor (IL-1R), as well as high activation of the nuclear factor-kB (NF-kB) transcription factors for pro-inflammatory cytokines [4]. Moreover, a condition of chronic low-grade inflammation may impair insulin sensitivity; insulin resistance worsens the inflammatory state, increasing abdominal obesity, intrahepatic fat stocking, vascular inflammation and endothelial dysfunction, which leads to an increase in cardiovascular risk [3].

Although calorie excess is the major determinant of weight gain, dietary fats have also been blamed for being the culprit in the dramatic increase in obesity and its associated diseases over the past half century. Over recent decades, dietary fat consumption has been discouraged but, in spite of the reduction in total daily individual fat intake by 10% and the increase in consumption of low-fat food, obesity rates are dramatically growing [5]. Furthermore, indistinctly cutting total fat intake from the diet inevitably leads to an increase in the consumption of highly processed grains and simple sugars while lowering the intake of liposoluble vitamins and unsaturated fats, especially from nuts, vegetable oils and fatty fish, which are particularly valuable for health [6].

Recent studies agree on the reduction of trans fatty acids (TFAs) derived from industrial foods and support the relevance of the type of fat consumed rather than total fat intake [7]. In fact, a recent meta-analysis of prospective cohort studies found no relationship between dietary total fat, polyunsaturated fatty acid (PUFA), monounsaturated fatty acid (MUFA) and saturated fatty acid (SFA) intake and cardiovascular disease (CVD) risk. Only trans fatty acid (TFA) intake had a dose–response association with CVD risk, while an inverse association between PUFA intake and CVD risk was found only among studies including a 10 year follow-up or more [8].

Although dietary SFAs are generally considered harmful to global health, these recent findings are quite controversial [9]. A possible explanation for these controversial results may derive from the type of SFA consumed in the diet. The absorption of short-chain fatty acids (SCFAs, two to six carbons) and medium-chain fatty acids (MCFAs, 8 to 12 carbons) occurs directly via portal circulation, while long-chain fatty acids (LCFAs, 14 to 20 or more carbons) are packaged into micelles and circulate via lymphatic-forming chylomicrons. While meat products are typically rich in LCFAs (i.e., palmitic and stearic acids), SCFAs (i.e., propionic acid and butyric acid) are produced when dietary fiber is fermented in the colon and are naturally present in milk and whole dairy products [10]. Scientific evidence from observational studies shows that higher intake of dairy products [11] and whole grains (rich in fiber) [12] are associated with a lower risk of CVD, while higher consumption of meat (red or processed) [13] is associated with higher risk. The evidence further suggests that the effects of SFA on lipid markers depend on the number of carbon atoms in the chain; a higher intake of lauric acid increases HDL cholesterol and reduces the TC to HDL ratio, while stearic, myristic and palmitic acids may raise LDL cholesterol [14].

Dietary fats may also have an effect on inflammation, which has a central role in the onset of many chronic pathologies. The first mechanism by which fats can affect inflammatory status is promotion of the translocation of microbial endotoxins, especially lipopolysaccharide (LPS), from the gut into the bloodstream [15]. LPS is a toxic cell-wall component of all our gut microbiome Gram-negative bacteria. The signaling of LPS is mediated through TLR4 receptors and leads to the stimulation of NF-kB, which in turn determines the secretions of many proinflammatory cytokines, such as IL-1, TNF-α, IL-6 and IL-8. SFAs seem to greatly stimulate inflammatory response because they are also structural components of LPS.

In contrast, PUFAs are known for their many beneficial properties, including the amelioration of neurologic and cardiovascular health and reduction of inflammation [16]. While evidence regarding n-6 PUFAs is still contrasting [17], n-3 PUFAs are precursors of lipid mediators, such as resolvins, docosatrienes and protectins, which have anti-inflammatory activities promoting neutrophil apoptosis and monocyte recruitment. n-3 PUFAs also affect lipid microdomains in cell membranes, playing a function in immune cell signaling pathways critical to inflammatory status [9,18].

Published studies on dietary fats generally consider overall consumption or, more recently, major subgroups. However, data on specific fatty acids are somewhat scarce, but they may represent important information for a better understanding of the role of dietary fats in human health. Thus, the objective of this study was to test whether specific types and subtypes of dietary fats are associated with metabolic outcomes in a cohort of adults from the south of Italy.

## 2. Materials and Methods

### 2.1. Study Population

The Mediterranean Healthy Eating, Aging and Lifestyle (MEAL) study is an observational study aimed at examining the relationship between lifestyle and dietary habits in the Mediterranean region and NCDs. The original cohort included a sample of 2044 randomized adults (≥18 years old) from the main districts of Catania, a city in the south of Italy. The recruitment and data collection were carried out between 2014 and 2015. Details of the project protocol are published elsewhere. All subjects involved in the study were advised about the objective of the project and provided written informed consent. All the study methods were carried out following the Declaration of Helsinki (1989). The study protocol was examined and accepted by the relevant ethical committee.

### 2.2. Data Collection

Data were collected with a presence-assisted interview through tablet computers. All subjects were supplied with a paper copy of the questionnaire in order to see each response option. However, the final answers were recorded instantly by the interviewer. The demographic information, such as age at recruitment, gender, highest educational level achieved, occupation (the most relevant job during the year before the investigation) or last occupation before retirement and marital status, were collected. Occupational status was classified as: (i) unemployed, (ii) low (unskilled workers), (iii) medium (partially skilled workers) and (iv) high (skilled workers). Educational status was categorized as: (i) low (primary/secondary), (ii) medium (high school) and (iii) high (university). The International Physical Activity Questionnaire (IPAQ) was used to examine motor activity [19], and it included a panel of questionnaires (five domains) investigating the time spent being physically active in the last week. In accordance with the IPAQ, physical activity level was classified as: (i) low, (ii) moderate and (iii) high. Smoking status was categorized as: (i) non-smoker, (ii) ex-smoker and (iii) current smoker, and alcohol consumption was classified as: (i) none, (ii) moderate drinker (0.1–12 g/d) and (iii) regular drinker (>12 g/d).

### 2.3. Cardio-Metabolic Outcomes

Anthropometric measurements were obtained following standard protocols [20]. Individuals were grouped according to body mass index (BMI) cut-offs as under/normal weight (BMI < 25 kg/m^2^), overweight (from BMI 25 to 29.9 kg/m^2^) and obese (BMI ≥ 30 kg/m^2^). Arterial blood pressure was measured in sitting position and after at least 5 min of rest at the end of the physical examination. Due to the possibility of differences in blood pressure measurement, the measurements were taken three times from the right arm, relaxed and well-supported by a table, with an angle of 45° from the trunk. The mean of the last two measurements was considered for inclusion in the database. Patients were considered hypertensive when average systolic/diastolic blood pressure levels were equal to or more than 140/90 mmHg, in accordance with the European Society of Cardiology (ESC)/European Society of Hypertension (ESH) guidelines, or when participants had been previously diagnosed with hypertension. Patients were considered dyslipidemic or diabetic if diagnosed by a physician with hypercholesterolemia/hypertriglyceridemia or diabetes, respectively. Previous diagnosis of diseases was collected from the medical records of the referred general practitioner.

### 2.4. Dietary Assessment

The nutritional assessment was conducted using two food frequency questionnaires (FFQs; a long and a short version) previously validated for the inhabitants of Sicily, south Italy [21,22]. The determination of the food ingested, the calories introduced and, especially, macro- and micro-nutrient intake were achieved through comparison with the food composition tables of the Research Center for Foods and Nutrition. The mean daily intake of each food was calculated in g or ml by considering the portion sizes provided in the FFQs and then converted to 24 h intake. Then, the content of total and specific fatty acids in each food was searched in the food composition tables of the Research Center for Foods and Nutrition and an estimation of their daily intake was calculated by multiplying the content of total and individual fatty acid molecules by the daily consumption of each food.

### 2.5. Statistical Analysis

Continuous variables are expressed as means and standard deviations (SDs), while categorical variables are expressed as frequencies of occurrence and percentages. Individuals were grouped by quartiles of total fat intake and distributions of background characteristics were compared between groups. Differences were tested with the chi-squared test for categorical variables, ANOVA for continuous variables distributed normally and the Kruskall–Wallis test for variables distributed non-normally. Energy-adjusted and multivariate logistic regression models were used to test the association between fat consumption and cardio-metabolic outcomes; the multivariate model adjusted for all other background characteristics (physical activity, educational status, occupational status, smoking status, alcohol consumption, menopausal status) was performed to test whether the observed associations were independent from the aforementioned variables. All reported p-values were based on two-sided tests and compared to a significance level of 5%. SPSS 17 (SPSS Inc., Chicago, IL, USA) software was used for all the statistical calculations.

## 3. Results

A total of 1936 individuals were included in the analyses in this study. The mean total dietary fat consumption varied across quartiles, from an average of 36.2 g/d in the lowest quartile to 97.8 g/d in the highest one. The major food groups contributing to dietary fats are shown in Figure 1: cheese and sweets were the main contributors, together with pasta/bread and olive oil; meat, yogurt, butter and eggs contributed only marginally to the total fat content in the sample’s diet.

The main background features of the cohort, distributed by quartiles of total dietary fat intake, are presented in Table 1. Subjects in the fourth quartile of dietary fat intake were significantly younger and had medium adherence to the Mediterranean diet. Similarly, significant differences were observed in the distributions of educational level, smoking status and physical activity, but with no linear trend. No significant differences among quartiles of dietary fat intake were found when considering BMI.

The association between total and classes of dietary fats and metabolic outcomes is shown in Table 2. Multivariate-adjusted analysis revealed a significant inverse association between total dietary fats and hypertension (for Q4, odds ratio (OR) = 0.57, 95% CI: 0.35, 0.91). Moreover, individuals reporting moderate consumption of total fats were less likely to have diabetes (for Q3, 0.27, 95% CI: 0.12, 0.61). Among single classes of dietary fats, individuals in the highest quartile of SFA intake were less likely to have hypertension (OR = 0.55, 95% CI: 0.34, 0.89). Moreover, subjects reporting moderate intake of MUFA were less likely to have hypertension and diabetes (OR = 0.61, 95% CI: 0.42, 0.88 and OR = 0.47, 95% CI: 0.22, 0.97, respectively). No associations were found between PUFA and any of the investigated outcomes.

Table 3 shows the association between specific sub-classes of fats and metabolic outcomes. Interestingly, individuals with the highest intake of SCSFA–MCSFA were less likely to have dyslipidemia (OR = 0.43, 95% CI: 0.23, 0.82) and diabetes (OR = 0.25, 95% CI: 0.09, 0.72). Subjects reporting moderate intake of C14:0 and C18:0 showed inverse associations with having hypertension (for Q3, OR = 2.36, 95% CI: 1.26, 4.24 and OR = 0.28, 95% CI: 0.12, 0.64, respectively) and, for the latter, diabetes (for Q3, OR = 0.19, 95% CI: 0.04, 0.97). Among single MUFAs, C14:1 intake resulted in an inverse association, in a linear way, with hypertension (OR = 0.57, 95% CI: 0.37, 0.88). C18:1 resulted in inverse associations with both diabetes (for Q4, OR = 0.21, 95% CI: 0.07, 0.67) and hypertension (for Q4, OR = 0.52, 95% CI: 0.30, 0.92). Conversely, C20:1 was directly associated with dyslipidemia (for Q4, OR = 3.35, 95% CI: 1.33, 8.42), hypertension (for Q3, OR = 1.81, 95% CI: 1.11, 2.97) and diabetes (for Q3, OR = 2.71, 95% CI: 1.02, 7.18). Among single PUFAs, C18:2 intake resulted in inverse associations with hypertension (for Q4, OR = 0.33, 95% CI: 0.18, 0.60) and diabetes (for Q4, OR = 0.32, 95% CI: 0.10, 0.97). Individuals in the highest quartile of C20:5 intake and those in the second quartile of C22:6 intake were less likely to have hypertension (OR = 0.30, 95% CI: 0.10, 0.89 and OR = 0.33, 95% CI: 0.12, 0.92, respectively).

## 4. Discussion

In the current study, we investigated the relationship between dietary fat subtype intake and cardio-metabolic risk factors in a cohort of Mediterranean adults. Interestingly, total SFA consumption was not detrimentally associated with any cardio-metabolic outcomes; conversely, individuals who had higher intake of total SFA were less likely to have hypertension and those who specifically consumed more SCSFA–MCSFA were less likely to have dyslipidemia and diabetes. Although SFAs have been assumed to be the main nutritional risk factor for cardio-metabolic diseases, recent studies have provided new controversial and interesting evidence suggesting that the SFA–CVD relationship may not be as strong as initially thought. A recent systematic review by the Cochrane group reported that a reduction in the intake of SFAs induced a 17% lowering of the risk of cardiovascular disease and that the beneficial effects increase when SFAs are replaced with PUFAs or starchy food [23]. However, cutting SFAs had a null effect on the other CVD end-points investigated and, furthermore, the putative adverse effect of SFAs on CVD events became non-significant when the analysis included only clinical trials that had successfully reduced SFA intake while removing those that were not successful [24]. The Prospective Urban Rural Epidemiological (PURE) study [25] conducted on 135,000 subjects without CVD demonstrated that increased consumption of total fats was linked with lower mortality and had a null association with CVD. Moreover, subjects in the highest quintile of SFA intake had a lower risk of stroke. Furthermore, a recent dose–response meta-analysis of cohort studies [8] highlighted that total fat, SFA, MUFA, and PUFA intake were not associated with CVD risk. Moreover, a meta-analysis of epidemiological studies conducted by SiriTarino [26] found no evidence that SFAs are associated with an increased risk of CHD. In addition, De Souza [27] and colleagues found no association between SFA intake and all-cause mortality, CHD, CHD mortality, ischemic stroke or type-2 diabetes among healthy subjects. A possible explanation for these controversial results on cardio-metabolic health may be the chain length of the fatty acids mainly consumed in diets [28]. SCFA may have a favorable impact on metabolism through activation of G protein-coupled receptors in endocrine and colon epithelial cells that, in turn, release anorectic hormones, such as glucagon-like peptide 1 (GLP) and peptide YY (PYY), which may contribute to reducing food intake and protecting individuals against obesity and diabetes, as shown in our results. Specifically, butyric acid (4:0) may have a beneficial effect on cardiovascular risk via inhibition of the NF-kB pathway and pro-inflammatory cytokines [29]. MCSFAs, such as caproic acid (6:0) and caprylic acid (8:0), have been shown in a cellular model to reduce the activity of fatty acid synthase (FAS), a primary enzyme of de novo lipogenesis that may contribute to the development of obesity and non-alcoholic fatty liver disease (NAFLD) [30]. Moreover, MCSFA may prevent endotoxic lipopolysaccharide (LPS)-mediated inflammation and lesions, which are linked to metabolic syndrome [31]. Caprylic acid (C8:0) and capric acid (C10:0) may reduce intestinal bile acid reabsorption with a simultaneous increase in excretion, which, in turn, lower TC and LDL-C [32].

In our cohort, myristic acid (14:0) intake was not associated with dyslipidemia and diabetes, while we found a positive association with hypertension. Although this molecule has been reported to play a role in post-translational protein changes and pathways that regulate several metabolic processes [33], data from the literature on its actual effect on metabolic health are not univocal: some studies reported potential beneficial effects from increasing HDL-C, such as reducing triglycerides levels, improving long-chain omega-3 levels in plasma phospholipids [34] and obesity-associated insulin resistance [35] and increasing LDL-C and apoB levels [36]. Among LCSFAs, in our study, stearic acid (18:0) was associated with a lower risk of diabetes and hypertension. Palmitic acid (16:0) and stearic acid (18:0) have been shown to increase cardiovascular risk through worsening of the lipid profile and alteration of inflammatory response [37,38], although the effects of stearic acid are less widely agreed on [39]. Moreover, stearic acid undergoes conversion to MUFA oleic acid (18:1) through the hepatic enzyme stearoyl–CoA–desaturase, which may in part explain the extensive detrimental effects.

Despite the new nutritional scenario regarding the role of dietary fat in cardio-metabolic health, the American Heart Association recently remarked on the importance of limiting SFA and substituting it with PUFA to reduce cardiovascular risk [40]. The American Dietary Guidelines Advisory Committee showed strong evidence that replacing SFAs with PUFAs reduces the risk of CHD events and CVD mortality, but there is a paucity of evidence to establish whether this substitution affects the risk of stroke or heart failure [41]. Data from previous epidemiological studies [42] and clinical trials [43] have shown that replacing SFA with PUFA and MUFA reduces the risk of combined CVD events, mainly lowering LDL-C and reducing inflammation. However, a recent umbrella review showed that replacing SFA with PUFA does not convincingly reduce cardiovascular events or mortality, probably due to the original invalidity of the diet–heart hypothesis and numerous research biases [44]. In our study, neither total PUFA intake nor the individual PUFAs were associated with dyslipidemia. PUFAs have always been generally considered “healthy” fats. They are implicated in vascular function, cell membranes, the nervous system and inflammation, being precursors of eicosanoids and beneficial lipid mediators, such as resolvins, docosatrienes and protectins. Previous evidence suggested that *n*-3 and *n*-6 PUFAs could have contrasting roles in human health. Indeed, *n*-6 PUFAs, such as linoleic acid (LA, 18:2), were thought to have pro-inflammatory effects because they can be converted into arachidonic acid (AA, 20:4) with consequent production of pro-inflammatory eicosanoids and, at the same time, reduce the conversion of *n*-3 PUFA alpha-linolenic acid (ALA, 18:3) into eicosapentaenoic acid (EPA, 20:5) and/or docosahexaenoic acid (DHA, 22:6) by competing for the same enzymes [45]. This hypothesis has lost its strength based on the latest results from clinical studies on humans demonstrating that LA intake has little influence on conversion into AA, while, conversely, it may be converted into nitrosylated LA and 13-hydroxyoctadecadienoic acid, which show anti-inflammatory effects [46]. Interestingly, in our study, subjects reporting a higher intake of LA were less likely to be hypertensive in a linear way. The blood pressure-lowering effects of LA are probably due to changes in vasodilator prostaglandin metabolism [47]. Moreover, we found that the intake of EPA and DHA was inversely associated with hypertension. A recent dose–response meta-analysis showed that EPA and DHA may reduce the risk of CHD by lowering high blood pressure among people already diagnosed with hypertension [48]. These fatty acids may exert cardioprotective effects, mainly by reducing ventricular fibrillation, heart rate and platelet aggregation through various mechanisms, including improved synthesis of eicosanoids [49,50] and inhibition of the NF-kB pathway by acting on PPAR-gamma and certain G-protein-coupled receptors in macrophages and adipocytes, thus reducing the production of inflammatory cytokines [51]. Finally, PUFA may also influence the risk of type-2 diabetes by reducing insulin resistance, activating PPAR-alpha and suppressing sterol regulatory binding protein-1c (SREBP-1c) [52]. A recent meta-analysis of RCTs showed that substituting SFAs with PUFAs improved glycemia and insulin resistance [53]. However, in line with our results, not all subtypes of PUFA appear to have the same effects on type-2 diabetes risk. The latest data show that *n*-6 PUFAs, but not *n*-3 PUFAs, may improve homeostasis model assessment—insulin resistance (HOMA-IR), lowering insulin in healthy subjects [54]. Moreover, a recent study reported that high levels of LA, but not AR, are associated with a lower risk of becoming diabetic [55]. In our study, only subjects with the highest intake of LA were less likely to have diabetes while the association with other PUFA was null.

In addition to PUFAs, MUFAs have also been considered for replacing dietary SFAs, as highlighted by the latest American dietary guidelines, although evidence is weak. Although substitution of SFAs with MUFAs has been associated with a decrease in metabolic syndrome [56], findings from a meta-analysis of observational studies showed null results on risk of CVD [8,42]. Our results showed that individuals with higher intake of oleic acid (18:1) were less likely to have hypertension and diabetes, while no association was found for dyslipidemia. Typical sources of MUFAs in the Mediterranean region are nuts, which have been demonstrated to have potential benefits for metabolic disorders [57]. MUFAs may have anti-inflammatory and antioxidant properties, lowering endoplasmic reticulum stress, inhibiting NF-kB transcription factor and acting through AMP-activated protein kinase (AMPK) phosphorylation, and may also reduce the polarization of M1 macrophages to M2 macrophages [51]. However, recent published studies report a beneficial effect of extra virgin olive oil and, probably, its polyphenols rather than the simple intake of MUFAs [58]. When MUFAs were isocalorically replaced with a non-lipidic component, such as carbohydrates, observed effects on blood lipids were rather small or negligible, and a recent review even highlighted the potentially negative effects of olive oil and oleic acid if introduced in large quantities [59]. These findings are in line with those of our study concerning gadoleic acid (20:1), a long-chain MUFA that appeared to be detrimentally associated with each investigated outcome in our study. The Ludwigshafen Risk and Cardiovascular Health Study showed an inverse association of gadoleic acid with LDL-C, HDL-C and eGFR but direct correlations with markers of inflammation and endothelial activation, as well as heart failure [60]. In contrast, a randomized controlled trial found that the supplementation of LCMUFAs (gadoleic acid and cetoleic acid) improved endothelial function by lowering trimethylamine-N-oxide levels, IL-6 and TNF-α, possibly due to improved gut microbiota profile [61]. The debate on the effects of LCMUFAs (more than 18 carbons) remains inconclusive as research is still scarce.

The results of the current study should be considered in light of several limitations. First, the observational nature of this investigation did not make it possible to define a causal relation between variables but only an association. The cross-sectional design of the study may have limited the interpretation of the results, as they may have suffered from revere causation. Moreover, although we performed multivariate-adjusted logistic regression analyses, residual vulnerability to type-1 errors still exists and should be taken into account. Another limitation concerns the dietary assessment method: although there is a univocal and perfect approach to collecting dietary data (with integrated use of multiple 24 h recalls and dietary diaries being highly desirable), FFQs are known to potentially under- or overestimate food intake due to recall bias, portion size miscalculation and social desirability bias. Finally, cases were confirmed by medical visit, but potential undiagnosed patients may have led to inclusion of false-negative cases in the study sample. These findings require further investigation in studies that are better designed, more controlled and use a prospective approach.

## 5. Conclusions

According to this new evidence, the debate on the role of fat in cardio-metabolic health remains open. In fact, the original diet–heart hypothesis seems to have lost its strength, while studies of the effects of saturated fat on health are in-depth, differentiating them according to the length of the carbon chain matrix and the global dietary pattern.

## Figures and Tables

**Figure 1 nutrients-14-04294-f001:**
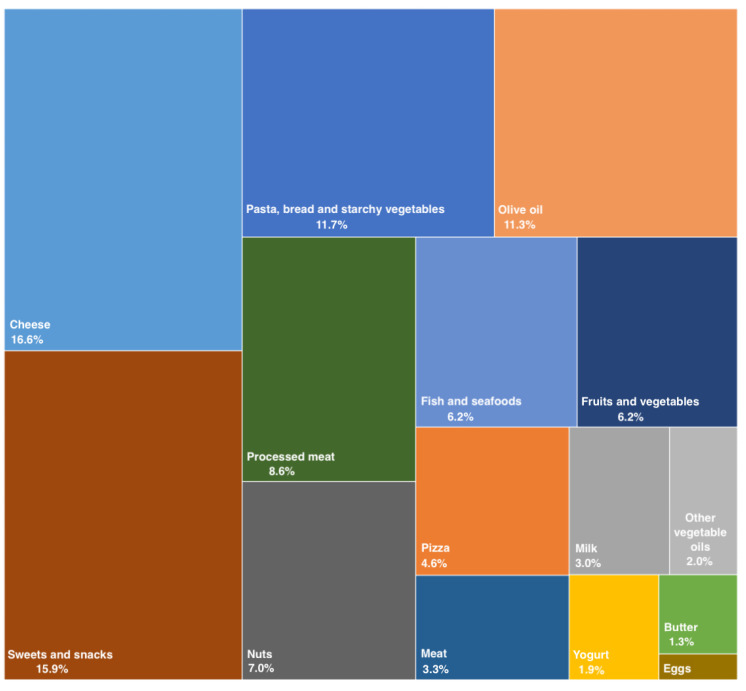
Major food groups contributing to the total dietary fat intake in the study sample. The size of the box indicates the actual contribution of the food group to total fat intake.

**Table 1 nutrients-14-04294-t001:** Background characteristics of the study sample and consumption of total dietary fats.

	Total Fats				*p*-Value
	Q1 (*n* = 448)	Q2 (*n* = 496)	Q3 (*n* = 506)	Q4 (*n* = 486)	
Total fats (g), mean (SD)	36.2 (4.8)	49.7 (3.9)	64.3 (4.3)	97.8 (31.6)	
Male	181 (40.4)	219 (44.2)	212 (41.9)	192 (29.5)	0.477
Female	267 (59.6)	277 (55.8)	294 (58.1)	294 (60.5)	
**Age, mean (SD)**	49.2 (18.2)	50.0 (17.5)	49.0 (17.8)	45.6 (16.8)	0.001
**Educational level, *n* (%)**					
Low	152 (33.9)	171 (34.5)	169 (33.4)	205 (42.2)	<0.001
Medium	146 (32.6)	193 (38.9)	211 (41.7)	170 (35.0)	
High	150 (33.5)	132 (26.6)	126 (24.9)	111 (22.8)	
**Smoking status, *n* (%)**					
Non-smoker	309 (69.0)	309 (62.3)	290 (57.3)	287 (59.1)	0.001
Current smoker	96 (21.4)	125 (25.2)	129 (25.5)	115 (23.7)	
Former smoker	43 (9.6)	62 (12.5)	87 (17.2)	84 (17.2)	
**Physical activity level, *n* (%)**					
Low	87 (19.7)	83 (18.7)	54 (13.6)	105 (23.6)	<0.001
Medium	214 (48.4)	194 (43.8)	227 (57.0)	221 (49.7)	
High	141 (31.9)	166 (37.5)	117 (29.4)	119 (26.7)	
**BMI categories, *n* (%)**					
Normal	224 (51)	221 (46)	213 (45.8)	193 (46.6)	0.610
Overweight	137 (31.2)	172 (35.8)	169 (36.3)	152 (36.7)	
Obese	78 (17.8)	87 (18.1)	83 (17.8)	69 (16.7)	
**Mediterranean diet adherence, *n* (%)**					
Low	283 (63.2)	259 (52.2)	279 (55.1)	230 (47.3)	<0.001
Medium	140 (31.3)	172 (34.7)	181 (35.8)	198 (40.7)	
High	25 (5.6)	65 (13.1)	46 (9.1)	58 (11.9)	

**Table 2 nutrients-14-04294-t002:** Association between total and classes of dietary fats and metabolic outcomes in the study sample.

	OR (95% CI)			
	Q1	Q2	Q3	Q4
	** *Hypertension* **			
**Total fats**				
Energy-adjusted	1	1.01 (0.78, 1.32)	0.65 (0.49, 0.86)	0.51 (0.36, 0.73)
Multivariate-adjusted	1	0.92 (0.66, 1.29)	0.53 (0.36, 0.76)	0.57 (0.35, 0.91)
**SFA**				
Energy-adjusted	1	1.01 (0.78, 1.32)	0.73 (0.55, 0.97)	0.68 (0.48, 0.96)
Multivariate-adjusted	1	0.94 (0.68, 1.32)	0.66 (0.46, 0.97)	0.55 (0.34, 0.89)
**MUFA**				
Energy-adjusted	1	1.04 (0.80, 1.35)	0.68 (0.51, 0.91)	0.55 (0.38, 0.78)
Multivariate-adjusted	1	0.99 (0.72, 1.39)	0.61 (0.42, 0.88)	0.65 (0.41, 1.04)
**PUFA**				
Energy-adjusted	1	0.85 (0.65, 1.11)	0.86 (0.64, 1.15)	0.74 (0.51, 1.09)
Multivariate-adjusted	1	0.83 (0.60, 1.17)	0.83 (0.56, 1.22)	0.71 (0.43, 1.18)
	** *Type-2 diabetes* **			
**Total fats**				
Energy-adjusted	1	0.45 (0.27, 0.76)	0.53 (0.31, 0.89)	0.81 (0.44, 1.50)
Multivariate-adjusted	1	0.39 (0.21, 0.74)	0.27 (0.12, 0.61)	0.95 (0.41, 2.19)
**SFA**				
Energy-adjusted	1	0.46 (0.27, 0.77)	0.63 (0.38, 1.04)	0.70 (0.38, 1.30)
Multivariate-adjusted	1	0.57 (0.31, 1.07)	0.52 (0.26, 1.03)	0.49 (0.20, 1.18)
**MUFA**				
Energy-adjusted	1	0.40 (0.23, 0.68)	0.63 (0.38, 1.06)	0.86 (0.46, 1.59)
Multivariate-adjusted	1	0.30 (0.15, 0.58)	0.47 (0.22, 0.97)	0.79 (0.33, 1.86)
**PUFA**				
Energy-adjusted	1	0.99 (0.59, 1.65)	1.19 (0.70, 2.04)	0.87 (0.42, 1.81)
Multivariate-adjusted	1	1.14 (0.60, 2.17)	0.84 (0.41, 1.76)	0.47 (0.18, 1.24)
	** *Dyslipidemias* **			
**Total fats**				
Energy-adjusted	1	0.79 (0.57, 1.10)	0.69 (0.48, 0.99)	0.87 (0.56, 1.36)
Multivariate-adjusted	1	0.66 (0.44, 1.01)	0.54 (0.33, 0.87)	0.59 (0.33, 1.06)
**SFA**				
Energy-adjusted	1	0.84 (0.60, 1.17)	0.89 (0.62, 1.27)	0.97 (0.63, 1.50)
Multivariate-adjusted	1	0.79 (0.52, 1.21)	0.73 (0.46, 1.18)	0.66 (0.37, 1.19)
**MUFA**				
Energy-adjusted	1	0.96 (0.69, 1.34)	0.81 (0.56, 1.17)	0.79 (0.50, 1.25)
Multivariate-adjusted	1	0.88 (0.58, 1.34)	0.75 (0.46, 1.22)	0.60 (0.34, 1.09)
**PUFA**				
Energy-adjusted	1	0.91 (0.65, 1.28)	0.89 (0.61, 1.29)	1.05 (0.65, 1.70)
Multivariate-adjusted	1	1.10 (0.72, 1.69)	0.67 (0.41, 1.09)	0.69 (0.37, 1.28)

Multivariate analysis adjusted for age, sex, educational status, smoking status, physical activity level, energy (kcal/d) and adherence to the Mediterranean diet. Abbreviations: Monounsaturated fatty acids (MUFA); Polyunsaturated fatty acids (PUFA); Saturated fatty acids (SFA).

**Table 3 nutrients-14-04294-t003:** Associations between specific fats and metabolic outcomes in the study sample.

	OR (95% CI)				OR (95% CI)				OR (95% CI)			
	Q1	Q2	Q3	Q4	Q1	Q2	Q3	Q4	Q1	Q2	Q3	Q4
	*Hypertension*				*Type-2 Diabetes*				*Dyslipidemias*			
**SFA**												
C4-C10	1	0.98 (0.68, 1.42)	0.97 (0.62, 1.05)	1.42 (0.87, 2.30)	1	0.99 (0.49, 2.00)	0.61 (0.25, 1.50)	0.25 (0.09, 0.72)	1	0.69 (0.43, 1.12)	0.49 (0.27, 0.89)	0.43 (0.23, 0.82)
C12:0	1	1.35 (0.90, 2.01)	1.10 (0.68, 1.80)	1.05 (0.60, 1.86)	1	0.58 (0.26, 1.50)	0.82 (0.32, 2.09)	1.56 (0.54, 4.52)	1	0.81 (0.47, 1.38)	0.82 (0.44, 1.51)	0.80 (0.40, 1.60)
C14:0	1	1.76 (1.12, 2.76)	2.36 (1.26, 4.24)	1.62 (0.73, 3.60)	1	1.20 (0.52, 2.78)	2.63 (0.74, 9.04)	2.96 (0.64, 13.59)	1	1.06 (0.59, 1.91)	1.77 (0.79, 3.97)	1.47 (0.54, 4.00)
C16:0	1	0.80 (0.44, 1.45)	1.15 (0.50, 2.65)	1.09 (0.37, 3.19)	1	0.83 (0.28, 2.42)	0.92 (0.17, 4.88)	1.16 (0.14, 9.68)	1	0.66 (0.32, 1.39)	0.80 (0.28, 2.27)	0.60 (0.16, 2.28)
C18:0	1	0.69 (0.38, 1.26)	0.28 (0.12, 0.64)	0.41 (0.15, 1.12)	1	0.49 (0.16, 1.51)	0.19 (0.04, 0.97)	0.26 (0.03, 1.93)	1	1.29 (0.62, 2.71)	0.95 (0.34, 2.62)	1.67 (0.49, 5.75)
C20:0	1	0.81 (0.57, 1.14)	1.11 (0.75, 1.65)	0.59 (0.35, 1.01)	1	0.81 (0.43, 1.49)	1.29 (0.61, 2.73)	1.35 (0.45, 4.03)	1	0.88 (0.57, 1.34)	1.02 (0.61, 1.68)	0.80 (0.40, 1.60)
C22:0	1	1.12 (0.80, 1.57)	1.10 (0.76, 1.58)	0.64 (0.40, 1.01)	1	0.88 (0.48, 1.62)	0.86 (0.41, 1.79)	0.97 (0.36, 2.64)	1	0.68 (0.45, 1.04)	1.01 (0.63, 1.60)	0.54 (0.29, 1.01)
**MUFA**												
C14:1	1	0.67 (0.48, 0.93)	0.53 (0.36, 0.76)	0.57 (0.37, 0.88)	1	0.60 (0.31, 1.16)	0.99 (0.49, 1.98)	2.03 (0.91, 4.56)	1	1.13 (0.74, 1.73)	0.94 (0.58, 1.53)	1.47 (0.85, 2.56)
C16:1	1	1.27 (0.88, 1.85)	0.94 (0.58, 1.55)	1.28 (0.68, 2.40)	1	0.65 (0.30, 1.37)	0.95 (0.36, 2.51)	1.62 (0.48, 5.46)	1	0.65 (0.40, 1.05)	0.65 (0.35, 1.21)	0.83 (0.38, 1.81)
C18:1	1	0.96 (0.67, 1.38)	0.72 (0.45, 1.14)	0.52 (0.30, 0.92)	1	0.30 (0.15, 0.63)	0.20 (0.07, 0.53)	0.21 (0.07, 0.67)	1	0.91 (0.57, 1.46)	0.72 (0.38, 1.34)	0.54 (0.26, 1.14)
C20:1	1	1.01 (0.70, 1.47)	1.81 (1.11, 2.97)	1.20 (0.60, 2.41)	1	1.99 (0.92, 4.28)	2.71 (1.02, 7.18)	1.73 (0.40, 7.47)	1	1.22 (0.75, 1.99)	1.32 (0.70, 2.48)	3.35 (1.33, 8.42)
C22:1	1	1.36 (0.96, 1.95)	1.08 (0.67, 1.71)	0.86 (0.44, 1.68)	1	1.54 (0.73, 3.22)	0.62 (0.24, 1.59)	1.41 (0.34, 5.81)	1	1.29 (0.80, 2.07)	0.59 (0.32, 1.09)	0.50 (0.20, 1.24)
**PUFA**												
C18:2	1	0.63 (0.42, 0.94)	0.56 (0.35, 0.91)	0.33 (0.18, 0.60)	1	0.90 (0.43, 1.85)	0.49 (0.20, 1.25)	0.32 (0.10, 0.97)	1	0.93 (0.56, 1.54)	0.69 (0.37, 1.28)	0.61 (0.29, 1.29)
C18:3	1	1.13 (0.76, 1.67)	1.13 (0.70, 1.84)	1.38 (0.77, 2.46)	1	0.97 (0.47, 2.00)	1.15 (0.48, 2.76)	1.47 (0.55, 3.95)	1	0.89 (0.54, 1.47)	1.19 (0.65, 2.18)	0.93 (0.46, 1.87)
C20:4	1	0.93 (0.67, 1.29)	0.87 (0.60, 1.26)	1.14 (0.73, 1.78)	1	1.30 (0.70, 2.39)	1.23 (0.62, 2.45)	0.97 (0.43, 2.18)	1	0.85 (0.55, 1.31)	0.85 (0.52, 1.38)	1.21 (0.69, 2.11)
C20:5	1	0.80 (0.48, 1.33)	1.00 (0.47, 2.14)	0.30 (0.10, 0.89)	1	1.08 (0.40, 2.89)	0.40 (0.09, 1.78)	0.95 (0.10, 9.34)	1	0.76 (0.39, 1.50)	0.67 (0.24, 1.83)	1.05 (0.27, 4.13)
C22:6	1	0.33 (0.12, 0.92)	0.51 (0.21, 1.26)	0.60 (0.29, 1.24)	1	0.38 (0.04, 3.46)	0.55 (0.07, 4.16)	0.86 (0.15, 4.83)	1	0.64 (0.17, 2.38)	1.20 (0.38, 3.78)	0.79 (0.32, 1.94)

## Data Availability

The data that support the findings of this study are available upon reasonable request.

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
