# Peer review of "Dietary Fats and Cardio-Metabolic Outcomes in a Cohort of Italian Adults"

_nutrients, 2022, doi:10.3390/nu14204294_

Round 1

Reviewer 1 Report

The authors present a novel cross-sectional analysis of the association between self-report dietary fat intake and cardio-metabolic outcomes. However, more information on the methods is needed, and the discussion needs editing to help with the readability. Below are my comments:

Abstract and title

·       The title and abstract should include cardio-metabolic instead of metabolic outcomes because both were assessed.

·       Provide methods for how cardio-metabolic outcomes were measured in the abstract

·       Provide an example of the chain length. E.g., The intake of fatty acids was also further examined according to the carbon chain length of each individual class (C14:1…).

Intro

·       grammatical errors need correcting

Methods

·       I suggest separating the cardio-metabolic outcomes from the data collection section into a separate methods section as they are the primary outcomes. Also need information on the methods for how Type-2 diabetes and Dyslipidemias were assessed. If these were just self report, this needs to be discussed as a limitation 

·       Additional information is needed about how dietary fat intake was extracted from the FFQ data. This will ensure the methods can be reproduced by other studies.

·       It might be helpful to include more detail on what fats will be examined and provide some examples of foods they are found in.

Results

·       Results are a bit dense and hard to follow, the information presented in the tables should not be repeated in the text.

Discussion

·       The discussion is quite long, and there are big paragraphs of text. Overall, it needs editing and cutting back to help with the flow.

·       In the discussion, fats are sometimes referred to by their name and sometimes by their carbon length. It is important that this is kept consistent.

·       Were multiple comparisons corrected in the analysis? If not, this needs to be discussed as a limitation.

·       Limitations need expanding upon with suggestions for future research.

Author Response

The authors present a novel cross-sectional analysis of the association between self-report dietary fat intake and cardio-metabolic outcomes.

Authors response: We would like to thank the Reviewer for the valuable comments on our manuscript.

However, more information on the methods is needed, and the discussion needs editing to help with the readability. Below are my comments:

Abstract and title

  •       The title and abstract should include cardio-metabolic instead of metabolic outcomes because both were assessed.

Authors response: Thank you for the suggestion. We corrected the title as requested.

  •       Provide methods for how cardio-metabolic outcomes were measured in the abstract

Authors response: we provided this information in the abstract

  •       Provide an example of the chain length. E.g., The intake of fatty acids was also further examined according to the carbon chain length of each individual class (C14:1…).

Authors response: Thank you for the suggestion. We listed each fatty acid with the number of carbons in the chain as requested.

Intro

  •       grammatical errors need correcting

Authors response: we revised the manuscript in context of grammatical errors.

Methods

  •       I suggest separating the cardio-metabolic outcomes from the data collection section into a separate methods section as they are the primary outcomes. 

Also need information on the methods for how Type-2 diabetes and Dyslipidemias were assessed. If these were just self report, this needs to be discussed as a limitation 

Authors response: Patients were considered dyslipidemic or diabetic only if diagnosed by a physician with hypercholesterolemia/hypertriglyceridemia or diabetes, respectively. Although it is not a self-report, potential undiagnosed patients may have led to inclusion of false negative cases into the study sample. We add this information in methods and at the end of the paper in limitations of the study. We have also separated cardio-metabolic outcomes from the data collection section as advised.

  •       Additional information is needed about how dietary fat intake was extracted from the FFQ data. This will ensure the methods can be reproduced by other studies.

Authors response: we agree with the reviewer, the description of the estimation of FA intake has been expanded. 

  •       It might be helpful to include more detail on what fats will be examined and provide some examples of foods they are found in. 

Authors response: We agree with the reviewer suggestion and we added a figure providing such information

Results

  •       Results are a bit dense and hard to follow, the information presented in the tables should not be repeated in the text.

Authors response: we agree with the reviewer, we have reformulated some sentences highlighting the most significant results

Discussion

  •       The discussion is quite long, and there are big paragraphs of text. Overall, it needs editing and cutting back to help with the flow.

Authors response: thank you for the suggestion we have revised and reduced the discussion.

  •       In the discussion, fats are sometimes referred to by their name and sometimes by their carbon length. It is important that this is kept consistent.

Authors response: thank you for the suggestion, we revised the paper so that the first time the fats appear in the text both the name and the length of the carbon chain are listed, while then only the name

  •       Were multiple comparisons corrected in the analysis? If not, this needs to be discussed as a limitation.

Authors response: yes, all analyses in Table 2 and in Table 3 are corrected for the main background characteristics.

  •       Limitations need expanding upon with suggestions for future research.

Authors response: we agree with the reviewer, we improved the limitation paragraph

Reviewer 2 Report

The study is a nice piece of work investigating the relationship of FA intake and CVD risks. Although the results are as elusive as previous literature showed, the work is clearly of high quality and worth of publication. Also, the manuscript is mostly well written. Several issues need some attention as follows:

Major points:

The conversion of food intake into FA intake is one of the first important steps for the assessment. Is it possible to compile a table which includes the majority of the food consumed and their FA contribution?

Some minor points:

Please add line numbers for the ease of review process.

1.       Abstract:

Regarding PUFA, C18:1 was inversely associated with diabetes (OR= 0.21, 95% CI: 0.07, 0.67) and hypertension (OR = 0.52, 95% CI: 0.30, 0.92), while C14:1 intake resulted inversely associated only with hypertension (OR = 0.57, 95%CI: 0.37, 0.88).

Clearly, 18:1 and 14:1 are not PUFA.

2.       Evidence further suggests that the effects of SFA on lipid markers depends also by the number of carbon chain; higher intake of lauric acid leads to an increase in HDL cholesterol and a reduction in the TC to HDL ratio while lauric, myristic and palmitic acids may raise LDL cholesterol [14].

Putting lauric acid before and after a “while” is not logically sound. Please rewrite this sentence.

3.       Spell out OR and CHD for its first appearance.

4.       Please clearly define Q1-Q4 quartiles.

5.       In addition to PUFA, MUFAs have also been considered into replacing dietary SFA as
highlighted by the latest American dietary guidelines although evidence is weak
corrected as “In addition to PUFA, MUFAs have also been considered for replacing dietary SFA as
highlighted by the latest American dietary guidelines although evidence is weak”.

Author Response

Reviewer 2

The study is a nice piece of work investigating the relationship of FA intake and CVD risks. Although the results are as elusive as previous literature showed, the work is clearly of high quality and worth of publication. Also, the manuscript is mostly well written. 

Authors response: We would like to thank the Reviewer for the valuable comments on our manuscript.

Several issues need some attention as follows:

Major points:

The conversion of food intake into FA intake is one of the first important steps for the assessment. Is it possible to compile a table which includes the majority of the food consumed and their FA contribution?

Authors response: we agree with the reviewer, the description of the estimation of FA intake has been expanded. 

Some minor points:

Please add line numbers for the ease of review process.

Authors response: this depends on the Editorial team (they provide a ready template to fill in)

  1.       Abstract:

“Regarding PUFA, C18:1 was inversely associated with diabetes (OR= 0.21, 95% CI: 0.07, 0.67) and hypertension (OR = 0.52, 95% CI: 0.30, 0.92), while C14:1 intake resulted inversely associated only with hypertension (OR = 0.57, 95%CI: 0.37, 0.88).“

Clearly, 18:1 and 14:1 are not PUFA.

Authors response: Thanks you for the suggestion we corrected as "Regarding PUFA, C18:2 and 20:5 were inversely associated with hypertension (OR = 0.33, 95% CI: 0.18, 0.60 and OR = 0.30, 95% CI: 0.10, 0.89, respectively)."

  1.       Evidence further suggests that the effects of SFA on lipid markers depends also by the number of carbon chain; higher intake of lauric acid leads to an increase in HDL cholesterol and a reduction in the TC to HDL ratio while lauric, myristic and palmitic acids may raise LDL cholesterol [14].Putting lauric acid before and after a “while” is not logically sound. Please rewrite this sentence.

Authors response: Thanks you for the suggestion we corrected as “Evidence further suggests that the effects of SFA on lipid markers depends also by the number of carbon chain; higher intake of lauric acid leads to an increase in HDL cholesterol and a reduction in the TC to HDL ratio while stearic, myristic and palmitic acids may raise LDL cholesterol”

  1.       Spell out OR and CHD for its first appearance.

Authors response: We spell both (OR in the abstract) and CHD at page 9

  1.       Please clearly define Q1-Q4 quartiles.

Authors response: thanks for the note, we added the mean fat content in each quartile in the table.

  1.       “In addition to PUFA, MUFAs have also been considered into replacing dietary SFA as

highlighted by the latest American dietary guidelines although evidence is weak“ corrected as “In addition to PUFA, MUFAs have also been considered for replacing dietary SFA as highlighted by the latest American dietary guidelines although evidence is weak”.

Authors response: Thanks you for the suggestion we corrected as advised

Round 2

Reviewer 1 Report

The authors have made substantial revisions to the manuscript. Well done. Below are my additional comments:

Abstract and title:

·       Please add a full stop to the end of this sentence. “However multiple systematic reviews and recent meta-analyses do not support the association between SFA and cardiovascular diseases”

·       In the abstract, Cases of hypertension were collected from previous doctor-confirmed diagnosis records, however, in the methods, Arterial blood pressure was assessed during the physical exam. Were both measures used? This needs some further clarification

Intro:

·       I think it is important to include a statement in the intro that weight gain and caloric intake are not the only risk factors for inflammation and cardiometabolic diseases, just part of the complex puzzle of risk factors.

Methods:

·       Please specify if medical records or self-reported diagnoses were used for cardio-metabolic outcomes.

Results:

·       The addition of the figure is helpful. However, some more information is needed to help interpret the figure. What does the size/colour of the box mean? Can you include labels or a key?

·       Please include the direction of the association when describing results; for example, what direction was the association between total dietary fats and hypertension?

·       There are grammatical errors that need editing. For example, “C18:1 resulted inversely associated in a linear way (OR= 0.21, 95% CI: 0.07, 0.67) with diabetes but only the highest quartile” this sentence needs rewording.

Discussion

·       Discussion still includes large chunks of text. It would help with the flow if these were broken down into smaller paragraphs.

·       Which statistical analysis is this statement referring to? “total SFA consumption was not associated with any cardio-metabolic outcomes”. This seems to be different to what was reported in the results: “Among single classes of dietary fats, individuals in the highest quartile (OR = 0.55, 95% CI: 0.34, 0.89) of saturated fat intake were less likely to have hypertension.”

  • Need to be consistent with terminology. For example, SFA vs saturated fat intake.
  • As many statistical analyses have been made in this current study, it is vulnerable to Type 1 error. If no corrections were made, such as the Bonferroni test, this should be included as a limitation of the study. Eg. The findings of this study should be interpreted with caution as no multiple-comparison corrections were made.
  • FFQ are not the gold standard dietary assessment: multiple 24hr diet recalls, and food diaries are less likely to be subject to recall bias and over and under-reporting. In addition, include a discussion about the need to assess the associations with biomarkers.
  • Keep the conclusion to findings from the current study in the context of the literature: “Moreover, the substitution of SFA with PUFA may not be beneficial to cardiovascular risk despite simply reducing total and low-density lipoprotein cholesterol.” This was not explored in the current study and would need to be further explored with randomised controlled trials.

Author Response

Reviewer 1

The authors have made substantial revisions to the manuscript. Well done. Below are my additional comments:

Authors response: We would like to thank the Reviewers for valuable comments and time dedicated to review our manuscript.

Abstract and title:

  •       Please add a full stop to the end of this sentence. “However multiple systematic reviews and recent meta-analyses do not support the association between SFA and cardiovascular diseases”

Authors response: we add a full stop at the end of the sentence as requested, thanks.

  •       In the abstract, Cases of hypertension were collected from previous doctor-confirmed diagnosis records, however, in the methods, Arterial blood pressure was assessed during the physical exam. Were both measures used? This needs some further clarification

Authors response: we collected both medical records and measured blood pressure. This is specified in the methods. We considered both measures.

Intro:

  •       I think it is important to include a statement in the intro that weight gain and caloric intake are not the only risk factors for inflammation and cardiometabolic diseases, just part of the complex puzzle of risk factors.

Authors response: Thank you for your suggestion. We added this statement: Recent evidence shows that the major risk factor for these conditions is chronic subclinical low-grade inflammation [2], that is usually determined for example by physical inactivity, tobacco, pollution, sleep alterations, dysbiosis, dysbiosis and infections (Furman et al. 2019). Among triggers for inflammation, poor diet and excessive weight play also an important role [3]

Methods:

  •       Please specify if medical records or self-reported diagnoses were used for cardio-metabolic outcomes.

Authors response: Thank you for the note, it was not clearly stated. When writing “previous diagnosis of..” we meant we collected medical records. We better specify this in the method section.

Results:

  •       The addition of the figure is helpful. However, some more information is needed to help interpret the figure. What does the size/colour of the box mean? Can you include labels or a key?

Authors response: size of the box means the contribution of a certain food group to the total fat intake. We also added the numerical reference within each box.

  •       Please include the direction of the association when describing results; for example, what direction was the association between total dietary fats and hypertension?

Authors response: we reported direction in all results.

  •       There are grammatical errors that need editing. For example, “C18:1 resulted inversely associated in a linear way (OR= 0.21, 95% CI: 0.07, 0.67) with diabetes but only the highest quartile” this sentence needs rewording.

Authors response: we corrected the sentence

Discussion

  •       Discussion still includes large chunks of text. It would help with the flow if these were broken down into smaller paragraphs.

Authors response: we shortened the discussion and better grouped it into paragraphs.

  •       Which statistical analysis is this statement referring to? “total SFA consumption was not associated with any cardio-metabolic outcomes”. This seems to be different to what was reported in the results: “Among single classes of dietary fats, individuals in the highest quartile (OR = 0.55, 95% CI: 0.34, 0.89) of saturated fat intake were less likely to have hypertension.”

Authors response: we meant “DETRIMENTALLY associated with any cardio-metabolic risk factors”, we corrected the sentence.

  • Need to be consistent with terminology. For example, SFA vs saturated fat intake.

Authors response: Thank you for your suggestion. We have replaced each name with its abbreviation

  • As many statistical analyses have been made in this current study, it is vulnerable to Type 1 error. If no corrections were made, such as the Bonferroni test, this should be included as a limitation of the study. Eg. The findings of this study should be interpreted with caution as no multiple-comparison corrections were made.

Authors response: we performed multivariate-adjusted logistic regression analyses, but we further added that residual vulnerability to type-1 error still exists and needs to be taken into account.

  • FFQ are not the gold standard dietary assessment: multiple 24hr diet recalls, and food diaries are less likely to be subject to recall bias and over and under-reporting. In addition, include a discussion about the need to assess the associations with biomarkers.

Authors response: we further added this limitation in the paragraph.

  • Keep the conclusion to findings from the current study in the context of the literature: “Moreover, the substitution of SFA with PUFA may not be beneficial to cardiovascular risk despite simply reducing total and low-density lipoprotein cholesterol.” This was not explored in the current study and would need to be further explored with randomised controlled trials.

Authors response: Thank you for the suggestion. We removed the sentence and we added a comment about our significant results and future perspectives: 

“In our cohort the consumption of SFA does not seem to be harmful to cardio-metabolic health and, on the contrary, MCSFA-SCSFA may exert beneficial effects. Further randomized controlled clinical trials are needed to validate our results, considering also the food matrix and the global dietary pattern”

Reviewer 2 Report

The authors made significant efforts in improving the manuscript according to reviewer's comments. The added content greatly improved clarity that the manuscript is now easy to follow.

Author Response

Reviewer 2

The authors made significant efforts in improving the manuscript according to reviewer's comments. The added content greatly improved clarity that the manuscript is now easy to follow.

Authors response: We would like to thank the Reviewers for valuable comments and endorsing our manuscript.